



# Elucidate the Formation Mechanism of Particulate Nitrate Based on Direct Radical Observations in Yangtze River Delta summer 2019

*Tianyu Zhai[a], Keding Lu[a, b*], Haichao Wang[c], Shengrong Lou[d], Xiaorui Chen[a,f], Renzhi Hu[e], Yuanhang Zhang[a, b*]*

[a] State Key Joint Laboratory of Environmental Simulation and Pollution Control, College of Environmental Sciences and Engineering, Peking University, Beijing 100871, China.

[b] Collaborative Innovation Center of Atmospheric Environment and Equipment Technology, Nanjing University of Information Science & Technology, Nanjing 210044, China.

[c] School of Atmospheric Sciences, Sun Yat-sen University, Guangzhou 510275, China.

[d] State Environmental Protection Key Laboratory of Formation and Prevention of the Urban Air Complex, Shanghai Academy of Environmental Sciences, Shanghai, 200223, China.

[e] Key Laboratory of Environmental Optics and Technology, Anhui Institute of Optics and Fine Mechanics, Chinese Academy of Sciences, Hefei, 230031, China.

[f] Now at: Department of Civil and Environmental Engineering, The Hong Kong Polytechnic University, Hong Kong, China.

*Correspondence to:*

Keding Lu (k.lu@pku.edu.cn), Yuanhang Zhang (yhzhang@pku.edu.cn )

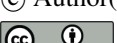



**Abstract.** Particulate nitrate ($NO_3^-$) is the one of the dominant components of fine particles in China, especially during pollution episodes, and has a significant impact on human health, air quality and climate. Here a comprehensive field campaign which focus on the atmospheric oxidation capacity and aerosol formation, and their effects in Yangtze River Delta (YRD) had been conducted from May to June, 2019 at a regional site in Changzhou, Jiangsu province in China. The concentration of $NO_3^-$, OH radical, $N_2O_5$, $NO_2$, $O_3$ and relevant parameters were measured simultaneously. We showed a high $NO_3^-$ mass concentration with $10.6 \pm 8.9$ μg m$^{-3}$ on average, which accounted for 38.3 % of water-soluble components and 32.0 % total $PM_{2.5}$, and followed by the proportion of sulfate, ammonium and chloride by 26.0 %, 18.0 % and 2.0 %, respectively. This result confirmed the heavy nitrate pollution in eastern China not only happened in winter but also summer time. High nitrate oxidation ratio (NOR) during this study emphasizes the strong atmospheric oxidation and fast nitrate formation capacity in YRD. It is found that $OH + NO_2$ at daytime dominates nitrate formation on clean days while $N_2O_5$ hydrolysis largely enhanced and become comparable with that of $OH + NO_2$ during polluted days (47.1 % and 52.9 %). An updated observed-constrain Empirical Kinetic Modeling Approach (EKMA) was used to assess the kinetic controlling factors of both local $O_3$ and $NO_3^-$ productions, which indicated that $O_3$-targeted scheme (VOCs:$NO_x$ = 2:1) is effective to mitigate the $O_3$ and nitrate pollution coordinately during summertime in this region. Our results promote the understanding of nitrate pollution mechanisms and mitigation based on field observation and model simulation, and call for more attentions to nitrate pollutions in summertime.

**Keywords:**

Nitrate pollution; Dinitrogen pentoxide; Nitrate formation; Pollution mitigation

**1 Introduction**

Chemical compositions of fine particle have been measured in China during past twenty



years and secondary inorganic aerosol is regarded as one of the dominant species in
aerosol (Cao et al., 2012; Hagler et al., 2006; Zhao et al., 2013; Andreae et al., 2008).
Since the Air Pollution Prevention and Control Action Plan, there has been a significant
decrease of $SO_2$, $NO_2$ and $PM_{2.5}$ concentration in China, while the inorganic nitrate
ratio in $PM_{2.5}$ increased and became the considerable component in $PM_{2.5}$ (Shang et al.,
2021; Zhang et al., 2022). Therefore, the comprehensive understanding of particlate
nitrate foramtion mechanism is essential and critical to mitigate haze pollution in China.
A massive research have been taken in China for investigating nitrate formation
mechanism and basic framework has been established (Sun et al., 2006; Chang et al.,
2018; Wu et al., 2019). In daytime, $NO_2$ + OH radical oxidation (Reaction 1) is the
major particulate nitrate formation pathway. The product ($HNO_3$) reacts with alkaline
substance in aerosol by which generating particulate nitrate. This pathway mainly
controled by precurors concentration as well as gas-particle partition of gaseous nitric
acid and particulate nitrate depends on temperature, relative humidity (RH), $NH_3$
concentration and aerosol acidity (Wang et al., 2009; Song and Carmichael, 2001;
Meng et al., 2020; Zhang et al., 2021). At night, $N_2O_5$ uptake is an important nitrate
formation pathway (Reaction 4)(Chen et al., 2020; Wang et al., 2022). $N_2O_5$ is formed
through $NO_2$ + $NO_3$ (Reaction 3) and there exsits a quick thermal equilibrium balance
($K_{eq} = 5.5 \times 10^{-17}$ $cm^{-3}$ $molecule^{-1}$ $s^{-1}$, 298 K). However, there are two problems remain
ambiguous in quantifying the contribution of $N_2O_5$ uptake to nitrate formation. The first
is the $N_2O_5$ heterogeneous uptake coefficent ($\gamma$) on ambient aerosol is highly varied with
the range from $10^{-4}$ to $10^{-1}$ based on previous lab and field measurments (Bertram and
Thornton, 2009; Brown et al., 2009; Wang et al., 2017c; Wang and Lu, 2016). The other
one is $ClNO_2$ production yield which inflences nitrate contribution duo to the large
variation range (Phillips et al., 2016; Staudt et al., 2019; Tham et al., 2018). Both the
two parameters are hardly to well-predicted by current schemes. Besides, $N_2O_5$
homogeneous hydrolysis, $NO_2$ heterogeneous uptake and $NO_3$ radical oxidation have
minor contribution to particulate nitrate under ambient condition (Brown et al., 2009;



Seinfeld and Pandis, 2016).

$$NO_2 + OH \rightarrow HNO_3 \qquad\qquad R1$$

$$NO_2 + O_3 \rightarrow NO_3 + O_2 \qquad\qquad R2$$

$$NO_2 + NO_3 + M \rightarrow N_2O_5 + M \qquad\qquad R3.1$$

$$N_2O_5 + M \rightarrow NO_2 + NO_3 + M \qquad\qquad R3.2$$

$$N_2O_5 + (H_2O \text{ or } Cl^-) \rightarrow (2 - \varphi) NO_3^- + \varphi ClNO_2 \qquad\qquad R4$$

As a key area of China's economy and industry, Yangtze River Delta (YRD) has
suffered severe air pollution during past decades and fine particle pollution in YRD has
raised a widespread concern (Guo et al., 2014; Zhang et al., 2015; Zhang et al., 2017;
Ming et al., 2017; Xue et al., 2019). However, most of these research focus on
wintertime $PM_{2.5}$ pollution and lack of measurements of critical intermediate species
and radicals to assess the importance of each nitrate formation pathway. In this study,
with the direct measurements of hydroxyl radical and the reactive nitrogen compounds
and chemical box model analysis, we explore the characteristics of nitrate and
precursors in YRD in the summer of 2019, the importance of particulate nitrate
formation pathways is quantified, and the impact factors are explored. Further
suggestion for summer pollution prevention and control for local area is proposed.
**2 Site description and methods**
**2.1 The campaign site**
This campaign had been taken place at a sub-urban sanatorium from May 30[th] to June
18[th] 2019 at Changzhou, China. Changzhou (119.95 °E, 31.79 °N) is located at Jiangsu
province and about 150 km northwest of Shanghai. The sanatorium which is located at
420 m east of Lake Ge (one of the largest lakes in Jiangsu province, 164 square
kilometers) is surrounded by farmland and fishpond. With the closest arterial traffic 1
km away, there are several industry zones 4 km to the east. The prevailing wind was



from south and south east sectors (about 30 % of the time) compared to 20 % from the
west sector, of which only 15 % came from the east. The wind speed was normally
lower than 5 m s$^{-1}$ with faster speed from the west. This site was influenced by both
anthropogenic and biological sources with occasionally biomass burning.

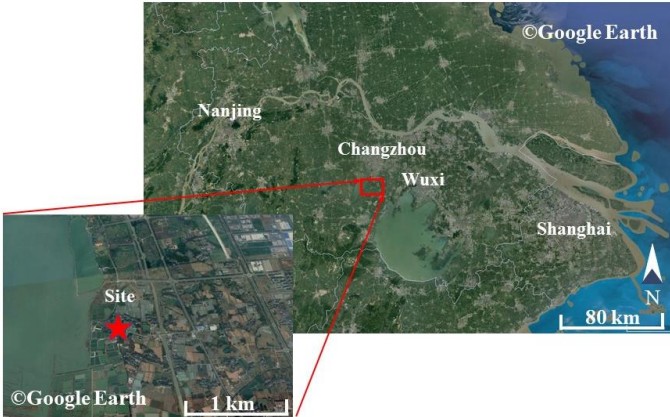

**Figure** 1 The location of campaign site (red star), Changzhou is located 150 km at the
northwest side of Shanghai.
**2.2 The instrumentation**
To comprehensive interpret the nocturnal atmospheric capacity and aerosol formation,
multiple gas and particle parameters were measured simultaneously and the related
instruments are listed in Table 1. $N_2O_5$ and Particle Number and Size Distribution
(PNSD) were measured on fourth floor of the sanatorium which is the top of the
building. Other instruments were set up in containers placed on the ground and 170 m
northeast of the building. These instruments monitored through the roof of containers
and inlets were circa 5 m above the ground.
$N_2O_5$ was measured by Cavity Enhanced Absorption Spectrometer (CEAS) based
on Lambert-Beer's law which was developed by (Wang et al., 2017b). Briefly, air
samples were drawn through the window and reached out of the wall 30 cm to prevent
influence from surface deposition. Aerosol membrane filter was deployed before



sample PFA tube and changed every 2 hours during the night to avoid a decrease in
$N_2O_5$ transmission efficiency due to the increased loss of $N_2O_5$ from the accumulated
aerosols on the filter. $N_2O_5$ was decomposed to $NO_3$ and $NO_2$ through preheating tube
which is heat at 130 °C and detected within a PFA-coated resonator cavity which is
heated at 110 °C to prevent the formation of $N_2O_5$ by reversible reaction subsequently.
At the end of each sampling cycle (5 min), 30 s injection of high concentration NO (10
ppm, 20 ml min$^{-1}$) which mixed with sample air was set to eliminate $NO_3$-$N_2O_5$ in the
system. The NO titration spectrums were adopted as dynamic background spectrum by
assuming that no $H_2O$ concentration variation in single sampling cycle. The loss of
$N_2O_5$ in the sampling system and filter were considered within data correction. The
limit of detection (LOD) was estimated to be 2.7 pptv (1 σ) with an uncertainty of 19 %.

OH radical measurement was conducted by Fluorescence Assay by Gas Expansion

Laser-Induced Fluorescence techniques (FAGE-LIF), ambient air was expanded
through a 0.4 mm nozzle to low pressure in a detection chamber, in where OH radical
irradiated by the 308 nm laser pulse at a repetition rate of 8.5 kHz (Chen et al., 2018).
$NO_x$ and $O_3$ were monitored by commercial monitors (Thermo-Fisher 42i and 49i).
Volatile organic compounds (VOCs) were measured by using automated Gas
Chromatograph equipped with a Mass Spectrometer and flame ionization detector (GC-
MS) with a time resolution of 60 min. The photolysis frequencies were determined from
the spectral actinic photon flux density measured by spectroradiometer (Bohn et al.,

2008).

$PM_{2.5}$ concentration was obtained by Tapered Element Oscillating Microbalance

(TEOM 1405, Thermo Scientific Inc). Aerosol surface concentration ($S_a$) was
converted from particle number and size distribution which measured by Scanning
Mobility Particle Sizer (SMPS, TSI 3936) and Aerosol Particle Sizer (APS, TSI 3321)
and modified to the wet particle-state $S_a$ with a hygroscopic growth factor (Liu et al.,
2013). The uncertainty of the wet $S_a$ was ~ 30 %. Meanwhile, water-soluble particulate
species as well as their gaseous precursors were analyzed through the Monitor for





AeRosols and GAses in ambient air (MARGA, Chen et al. (2017)). Meteorological data,
including the temperature, relative humidity (RH), pressure, wind speed, and wind
direction, were also available.
**Table 1** The observed gas and particle parameters during the campaign.

| Parameters | Detection of limit | Method | Accuracy |
|---|---|---|---|
| $N_2O_5$ | 2.7 pptv (1 σ, 1 min) | CEAS | ± 19 % |
| OH | $1.6 \times 10^5$ cm$^{-3}$ (1 σ, 60 s) | LIF[a] | ± 21 % |
| NO | 60 pptv (2 σ, 1 min) | PC[c] | ± 10 % |
| $NO_2$ | 0.3 ppbv (2 σ, 1 min) | PC[c] | ± 10 % |
| $O_3$ | 0.5 ppbv (2 σ, 1 min) | UV photometry | ± 5 % |
| VOCs | 20-300 pptv (60 min) | GC-MS | ± 15 % |
| $PM_{2.5}$ | 0.1 μg m$^{-3}$ (1 min) | TEOM[d] | ± 5 % |
| Photolysis frequencies | $5 \times 10^{-5}$ s$^{-1}$ (1 min) | SR[e] | ± 10 % |
| PNSD | 14 nm -700 nm (4 min) | SMPS, APS | ± 10 % |
| $HNO_3$, $NO_3$, HCl | 0.06 ppbv (30 min) | MARGA[f] | ± 20 % |
| $NH_4^+$, $NO_3^-$, $Cl^-$, $SO_4^{2-}$ | 0.05 μg m$^{-3}$ (30 min) | MARGA[f] | ± 20 % |

[a] Laser-induced fluorescence; [b] Chemiluminescence; [c] Photolytic converter; [d] Tapered
Element Oscilating Microbalance; [e] Spectroradiometer; [f] the Monitor for AeRosols and
GAses in ambient air.
**2.3 The empirical kinetic modelling approach**
A box-model coupled with the Regional Atmospheric Chemical Mechanism version 2
(RACM2, Goliff, Stockwell & Lawson, 2013) is used to conduct the mitigation
strategies studies. The model is operated in one-hour time resolution with measurement
results of temperature, relative humidity, pressure, CO, $NO_2$, $H_2O$, photolysis
frequencies and aggregated VOCs input to constrain the model. It should be noted that
HONO concentration is simply calculated by $NO_2$ times 0.02 which is suggested by
Elshorbany et al. (2012) and has been used in box model before (Lou et al., 2022).
Long-live species such as $H_2$ and $CH_4$ are set as constants (550 ppbv and 1900 ppbv
respectively). What's more, a 13-hour constant loss rate of unconstrained intermediate
and secondary products, which is the result of synthetic evaluating secondary
simulation of secondary species, is set for representing the multi-effects of deposition,
transformation and transportation.

The approaches of chemical production of $O_3$ (P($O_3$)) and inorganic nitrate



(P($NO_3^-$)) are using previously described expression (Tan et al., 2021; Tan et al., 2018)
in Equation 1 and 4:

$$P(O_3) = F(O_3) - D(O_3) \qquad \text{Eq1}$$

$$F(O_3) = k_{HO_2+NO}[NO][HO_2] + k_{(RO_2+NO)eff}[NO][RO_2] \qquad \text{Eq2}$$

$$D(O_3) = k_{OH+NO_2}[OH][NO_2] + (k_{OH+O_3}[OH] + k_{HO_2+O_3}[HO_2] + k_{alkenes+O_3}[alkenes])[O_3] \qquad \text{Eq3}$$

$$P(NO_3^-) = P(HNO_3) + P(pNO_3^-) \qquad \text{Eq4}$$

$$P(HNO_3) = k_{OH+NO_2}[OH][NO_2] \qquad \text{Eq5}$$

$$P(pNO_3^-) = 0.25(2 - \varphi) \, C \, \gamma \, S_a \, [N_2O_5] \qquad \text{Eq6}$$

briefly, P($O_3$) is net ozone production which calculated by peroxyl radial + NO
oxidation (Eq. 2) minus chemical loss of $O_3$ and $NO_2$ (Eq. 3). P($NO_3^-$) is constituted by
reaction OH + $NO_2$ (Eq. 5) and $N_2O_5$ heterogenous uptake (Eq. 6). Here, rate constants
of reactions are obtained from NASA JPL Publication (Burkholder et al., 2015) or
RACM2 (Goliff et al., 2013). $\gamma$ is the $N_2O_5$ uptake coefficient which is calculated from
parameterization ($\gamma_{\_P}$, more details in chapter 3.3). $\varphi$ represents $ClNO_2$ production yield
through $N_2O_5$ hydrolysis and the mean value reported in Xia et al. (2020) are used in
this work.
Empirical Kinetic Modeling Approach (EKMA) was innovated for studying the
effects of precursors VOCs, NOx reactivity on the region's ozone pollution by Kanaya
et al, which help recognize the region's susceptibility to precursors by weight and
become a prevalent tool to study the process of ozone formation (Tan et al., 2018; Yu
et al., 2020b; Kanaya et al., 2008). The prevention and control problem of pollutant
generation can be transformed through EKMA curve to reduce its precursors emissions.
Furthermore, the precursors reduction scheme needed for pollutant total control is given
qualitatively. P($NO_3^-$) can also be analyzed through EKMA for the nonlinear secondary
formation relationship with precursor reactivity. Here, isopleth diagram of the net ozone
production rate as functions of the reactivities of $NO_x$ and VOCs can be derived from
EKMA. In detail, 0.01 to 1.2 emission reduction strategy assumptions are exponential
interpolation into 20 kinds of emission situation of $NO_x$ and VOCs respectively, which
in total counts 400 scenarios.
**2.4 The calculation of aerosol liquid water content**
Aerosol liquid water content (ALWC) is calculated through ISORROPIA II





(Fountoukis and Nenes, 2007). Forward mode is applied in this study. Furthermore,
water-soluble ions in $PM_{2.5}$ and gaseous species ($NH_3$ + $HNO_3$ + $HCl$) obtained from
MARGA, along with RH and T are input as initial input. In addition, metastable aerosol
state is chosen since high RH during this campaign.

**3 Result and discussion**


**3.1 Overview of measurements**


The time used in this study is China Standard Time (UTC + 8) and the local sunrise and
sunset time during the campaign were around 5 am and 7 pm, respectively. The whole
campaign period is divided into the four $PM_{2.5}$ clean periods and four $PM_{2.5}$ polluted
periods (9 out of 14 days, latter polluted periods and day refer to $PM_{2.5}$ pollution except
specified description) according to the Chinese National Air Quality Standard
(CNAAQS) Grade I of daily $PM_{2.5}$ concentrations (< 35.0 $\mu g\ m^{-3}$). Figure 2 shows the
meteorological parameters, gas-phase and particulate species timeseries during the
observation. During the campaign, the temperature was high and the maximum reached
34.5 °C, with an average 25.1 ± 3.7 °C. RH changed drastically from 21 % to 88 %,
with mean value at 58.9 ± 14.0 %. Mean $NO_2$ concentration was 14.8 ± 9.5 ppbv.
Meanwhile, $O_3$ average was 54.6 ± 28.8 ppbv, exceeding CNAAQS Grade II for
maximum daily average 8 h ozone (160 $\mu g\ m^{-3}$) on 14 out of 19 days and exceeding
200 $\mu g\ m^{-3}$ on 6 days.

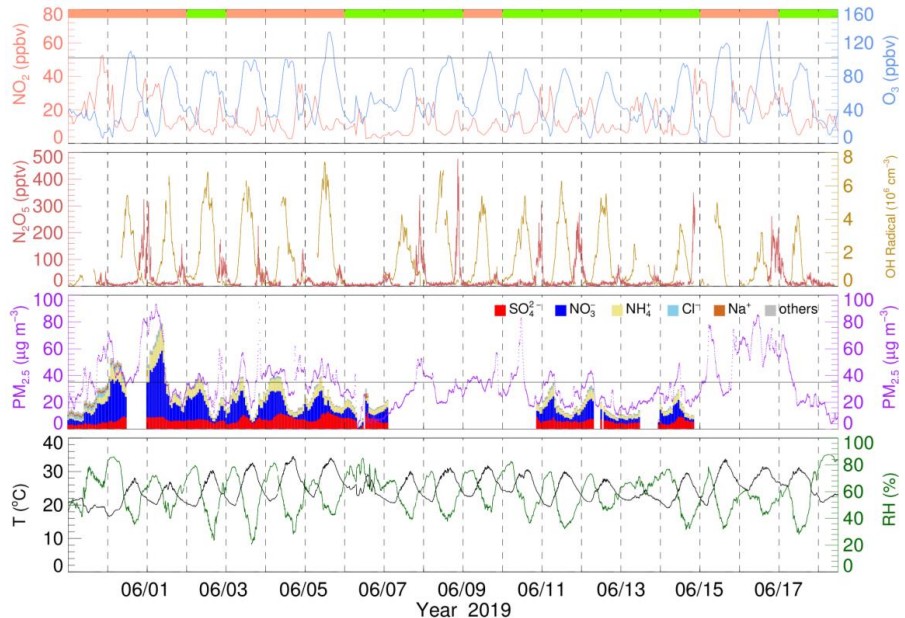


**Figure 2** Timeseries of $NO_2$, $O_3$, $N_2O_5$, OH radical, $PM_{2.5}$ and water-soluble particulate species, temperature and RH. The vertical dotted line represents zero clock. The black horizontal solid line in $O_3$ and $PM_{2.5}$ panel represents Chinese national air quality standard for $O_3$ and $PM_{2.5}$ respectively. Top panel color blocks represent $PM_{2.5}$ clean day (light green) and $PM_{2.5}$ polluted day(salmon) respectively.

217 Daytime OH radical ranged from $2 \times 10^6$ to $8 \times 10^6$ molecular $cm^{-3}$ with daily peak

218 over $3 \times 10^6$ molecular $cm^{-3}$. Maximum OH radical reached $8.18 \times 10^6$ molecular $cm^{-3}$

219 in this campaign. Comparing with other summertime OH radical observed campaign in

220 China, OH radical concentration in this site is relatively low but still on the same order

221 of magnitude (Lu et al., 2012; Lu et al., 2013; Ma et al., 2022; Tan et al., 2017;

222 Woodward-Massey et al., 2020; Yang et al., 2021). $N_2O_5$ mean concentration was 21.9

223 ± 39.8 pptv with nocturnal average 61.0 ± 63.1 pptv and daily maximum over 200 pptv

224 at 8 nights. The maximum concentration of $N_2O_5$ (477.2 pptv, 5 min resolution)

225 appeared at 20:47 June 8$^{th}$. The average $NO_3$ radical production rate $P(NO_3)$ is 2.1 ±

226 1.4 ppbv $h^{-1}$ with nocturnal average $P(NO_3)$ 2.8 ± 1.6 ppbv $h^{-1}$ and daytime $P(NO_3)$ 2.2

227 ± 1.4 ppbv $h^{-1}$. $P(NO_3)$ is about twice of documented value in Taizhou and North China

228 Plain (Wang et al., 2017a; Wang et al., 2018b; Wang et al., 2020a), but close to another





result in YRD before (Chen et al., 2019). Average of $PM_{2.5}$ was $34.6 \pm 17.8$ µg m$^{-3}$ with
maximum reach $163.0$ µg m$^{-3}$. The water-soluble components of $PM_{2.5}$ are displayed as
well, the average $NO_3^-$ concentration was $10.6$ µg m$^{-3}$, which accounts for 38.3 % mass
concentration of water-soluble components and 32.0 % total $PM_{2.5}$, while proportion of
sulfate, ammonium and chloride are 26.0 %, 18 % and 2.0 % respectively. To sum up,
during campaign period, the pollution of $PM_{2.5}$ would be exacerbated in general on high
$O_3$ and $NO_2$ days. Precipitation occurred during four clean processes receded pollutant
concentration, otherwise, the pollution condition remained severe.

The mean diurnal variations (MDC) of temperature, RH, $NO_2$, $O_3$, $P(NO_3)$, $N_2O_5$,

OH radical and $PM_{2.5}$ in different air quality are shown in Figure 3. The temperature,
RH and OH radical MDC show indistinctive difference between clean day (CD) and
polluted day (PD). The MDC of $NO_2$ has two concentration peaks appeal at 06:00 and
21:00 on CD, while at PD, its peak appeals at 20:00 and maintain high level during
whole night. $O_3$ diurnal pattern reflects a typical urban-influenced character with
maximum $O_3$ peak lasts four hours from 14:00 to 17:00 with polluted-day $O_3$ peak
concentration 1.2 higher than clean-day. $P(NO_3)$ grows after $O_3$ peak and maximum
$P(NO_3)$ shows at 19:00 with average value 1.7 ppbv h$^{-1}$ on clean day. By contrast, mean
polluted-day $P(NO_3)$ is 2.6 ppbv h$^{-1}$ and maximum value reach 4.7 ppbv h$^{-1}$. In contrast,
the clean-day $N_2O_5$ has higher average and maximum concentration than PD which
suggests faster removal process during PD. $PM_{2.5}$ have similar trend with $P(NO_3)$ and
has higher concentration during nighttime.

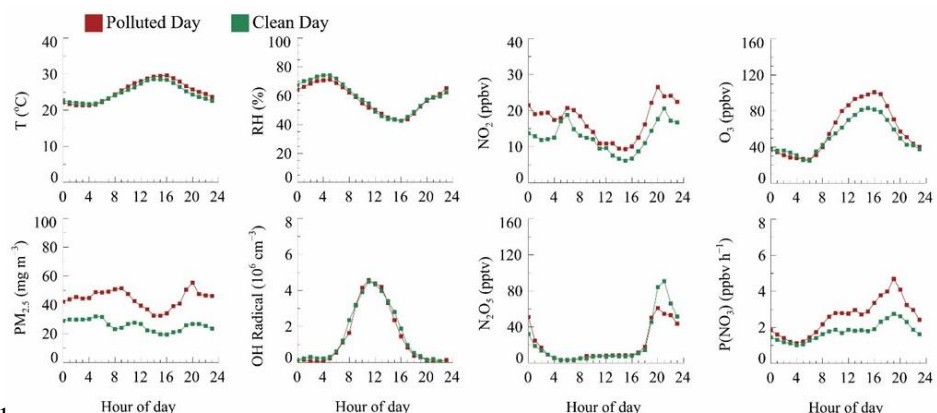

**Figure 3** The mean diurnal variations of temperature, RH, $NO_2$ (Salmon), $O_3$, $P(NO_3)$,
$N_2O_5$, OH radical(orange) and $PM_{2.5}$ of clean day and polluted day.
**3.2 The evolution of nitrate pollution**
Figure 4 (a) shows the relationship of nitrate and sulfate with water-soluble ion. Nitrate
has positive correlation with particulate water-soluble ion while sulfate ratio having
inverse correlation. With $PM_{2.5}$ concentration increasing, nitrate proportion increasing
rapidly and keep high weight at heavy $PM_{2.5}$ period while sulfate appears opposite
phenomenon. Once the mass concentration of water-soluble ion over 30 μg m$^{-3}$, the
mass fraction of nitrate in total water-soluble ion is up to 50 % on average. This result
illustrates that particulate nitrate is one of the vital sources of particulate matter
explosive growth.

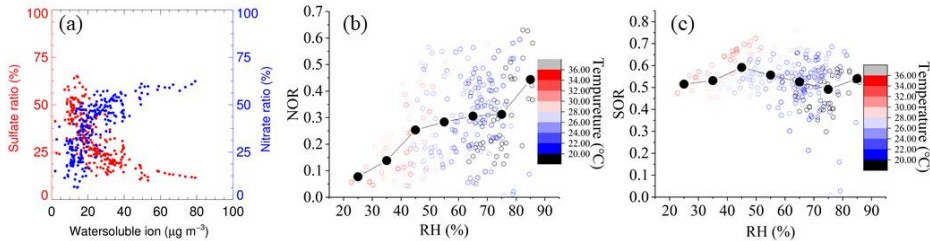

**Figure 4** (a) Particulate ion mass concentration ratio of nitrate and sulfate to water
soluble ion. (b) NOR against RH, colored with temperature. (c) SOR against RH,
colored with temperature.





To further assess the conversion capacity of nitrate and sulfate in this site, the sulfur

oxidation ratio (SOR) and the nitrogen oxidation ratio (NOR) are used for indicating
secondary transformation ratio of $SO_2$ and $NO_2$ respectively (Sun et al., 2006). SOR
and NOR are estimated using formulae below:

$$SOR = \frac{nSO_4^{2-}}{nSO_4^{2-}+nSO_2} \qquad\qquad Eq7$$

$$NOR = \frac{nNO_3^-}{nNO_3^-+nNO_2} \qquad\qquad Eq8$$

here n refers to the molar concentration. The higher SOR and NOR represent more
oxidation of gaseous species into secondary aerosol. As depicted in Figure 4 (b-c), NOR
rapid increases at RH < 45 %, remains constant at 45 % < RH < 75 % and ends with a
sharply increase at RH > 75 %. In addition, NOR has inverse correlation with
temperature which reflects the importance of nighttime secondary transformation and
the influence of negative correlation of gas-solid equilibrium between particulate nitrate
and gaseous $HNO_3$. During the study period, not only the average concentration of $NO_2$
is higher among PD, but also there is significate difference between PD and CD NOR.
The average values of NOR are 0.32 in PD, 0.25 in CD respectively which manifests
that the more secondary transformation and pollution potential in PD. On the contrast,
the SOR stays constant at high value (~ 0.5) during the whole RH scale which shows
different pattern with previously research (Li et al., 2017; Zheng et al., 2015). One
possible explanation is that $SO_2$ concentration stays low level during the whole
campaign (4.4 ± 2.4 ppbv on average) and $SO_2$ oxidation depends on limit of $SO_2$
instead of oxidation capability. Meanwhile, mean SOR in both situations are over 0.5
(0.52 in CD and 0.56 in PD), further supporting the $SO_2$ limited hypothesis. Besides,
Table 2 summaries NOR and SOR values in YRD. NOR and SOR in this study are
similar with values reported in other YRD research (Shu et al., 2019; Zhang et al.,
2020b; Qin et al., 2021; Zhao et al., 2022), except values in 2013 (Wang et al., 2016),





but higher than north China study (Cao et al., 2017) which emphasize the strong
atmospheric oxidation capacity in YRD region.
**Table 2** Statistical result of NOR and SOR in YRD

| Location and Year | SOR | | | | NOR | | | | References |
|---|---|---|---|---|---|---|---|---|---|
| | Max | Min | Mean | SD | Max | Min | Mean | SD | |
| Nanjing 2013 Winter | 0.42 | 0.10 | 0.28 | 0.11 | 0.29 | 0.15 | 0.21 | 0.05 | |
| Suzhou 2013 Winter | 0.41 | 0.15 | 0.27 | 0.11 | 0.30 | 0.06 | 0.16 | 0.08 | |
| Lin'an 2013 Winter | 0.50 | 0.19 | 0.35 | 0.11 | 0.24 | 0.12 | 0.18 | 0.05 | (Wang et al., 2016) |
| Hanghou 2013 Winter | 0.30 | 0.14 | 0.21 | 0.06 | 0.11 | 0.06 | 0.09 | 0.02 | |
| Ningbo 2013 Winter | 0.35 | 0.09 | 0.21 | 0.11 | 0.23 | 0.03 | 0.11 | 0.07 | |
| YRD 2016 Summer | - | - | 0.347 | - | - | - | 0.11 | - | (Shu et al., 2019) |
| YRD 2016 Winter | - | - | 0.247 | - | - | - | 0.15 | - | |
| Nanjing 2019 spring | 0.48 | 0.38 | - | - | 0.31 | 0.29 | - | - | (Qin et al., 2021) |
| Changzhou 2019 spring | 0.35 | 0.3 | - | - | 0.27 | 0.23 | - | - | |
| Changzhou 2019 Winter | 0. 68 | 0.24 | 0.35 | 0.12 | 0.44 | 0.13 | 0.2 | 0.1 | (Zhang et al., 2020b) |
| Changzhou 2019 Summer | 0.16 | 0.76 | 0.54 | 0.1 | 0.08 | 0.63 | 0.28 | 0.14 | This work |

**3.3 The derivation of N₂O₅ uptake coefficient**
Statistical analysis of observation above highlights the fast formation of particulate
nitrate. In order to assess the contribution of N₂O₅ hydrolysis to particular nitrate
formation, two methods are applied to calculate N₂O₅ uptake coefficient. The first
method is stationary-state approximation (Brown et al., 2003). By assuming that the
rates of production and loss of N₂O₅ are approximately in balance, the total loss rate of
N₂O₅ ($k_{N_2O_5}$) can be calculated through equation 9. The $k_{N_2O_5}$ is main dominated by
N₂O₅ heterogeneous uptake, since homogeneous hydrolysis of N₂O₅ contributes very
little (Brown and Stutz, 2012). N₂O₅ uptake coefficient through steady-state (note as
$\gamma\_s$) is derived as equation 10. Here C is the mean molecule speed of N₂O₅, $S_a$ is the
aerosol surface concentration.

$$\tau_{ss}(N_2O_5) = \frac{[N_2O_5]}{k_{R3.1}[NO_2][O_3]} = \left(k_{N_2O_5} + \frac{k_{NO_3}}{K_{eq}[NO_2]}\right)^{-1} \qquad \text{Eq9}$$

$$k_{N_2O_5} = 0.25\, C\, \gamma\_S\, S_a \qquad \text{Eq10}$$

Due to fast variety of NO₃ loss rate from VOCs, the steady-state method has been



unattainable in conditions affected by emission interferences. During the whole
campaign, we only retrieve three valid fitting results. As shown in Figure 5, the fitted
$\gamma\_s$ are ranged from 0.057 to 0.123, which are comparable with Taizhou (0.041, Wang
et al. (2020a)) and much higher than other results in China (Yu et al., 2020a; Wang et
al., 2018a; Wang et al., 2020b; Wang et al., 2017a). The calculated $k_{NO_3}$ ranged from
0.002 to 0.16 s$^{-1}$, represents drastic VOCs change during this campaign.

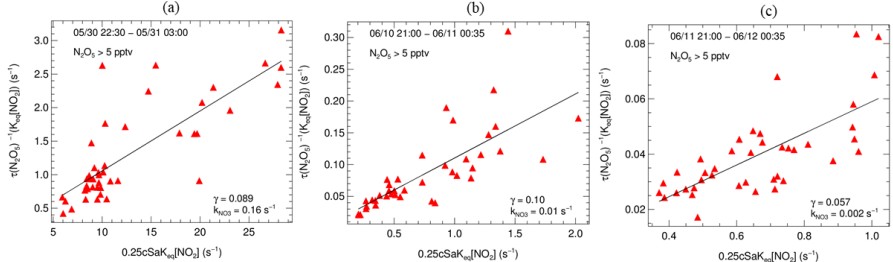


**Figure 5** Derived $N_2O_5$ uptake coefficients from $N_2O_5$ steady lifetime ($\gamma(N_2O_5\_S)$)
with $NO_2$ and $S_a$, plots (a-c) represent the linear fitting results at the nights of 05/30,
06/10 and 06/11, respectively.

The other approach is the parameterization by (Yu et al., 2020a) which depicted as

follows:

$$\gamma\_P = \frac{4}{c}\frac{V_a}{S_a} K_H \times 3.0 \times 10^4 \times [H_2O] \left(1 - \frac{1}{\left(0.033 \times \frac{[H_2O]}{[NO_3^-]}\right) + 1 + \left(3.4 \times \frac{[Cl^-]}{[NO_3^-]}\right)}\right) \qquad Eq11$$


where $V_a/S_a$ is the measured aerosol volume to surface area ratio by SMPS; $K_H$ is
Henry's law coefficient which is set as 51 as recommended; $[NO_3^-]$ and $[Cl^-]$ are aerosol
inorganic concentration measured by Marga; $[H_2O]$ is aerosol water content calculated
through ISORROPIA II. The parameterization calculated $N_2O_5$ uptake coefficient (note
as $\gamma\_P$) vary from 0.014 to 0.094 with average 0.035.

Furthermore, we compare the difference between $\gamma\_s$ and $\gamma\_P$. Taking the night of

May 30$^{th}$ as example, the $\gamma\_s$ is 0.10 while $\gamma\_P$ ranges from 0.021 to 0.037 with average
value as 0.026. The difference between steady-state and parameterization is significant.





### 3.4 Quantifying the contribution of nitrate formation pathways

After the $N_2O_5$ uptake coefficient counted, nitrate production potential ($P(NO_3^-)$) can be calculated. Here $N_2O_5$ uptake coefficient is set as 0.035 which is the average value derived from parameterization, while the production ratio of $NO_3^-$ (by considering $ClNO_2$ yield of 0.54) is set as 1.46 from former study (Xia et al., 2020). Gas particle distribution is considered by the result of particular nitrate and gas-phase nitrate by MARGA (input $HNO_3/NO_3^-$ ratio to the model as $OH + NO_2$ nitrate production rate). Subsequent discussion focuses on $OH + NO_2$ and $N_2O_5$ heterogeneous uptake.

The mean diurnal variations of nitrate production potential of clean and polluted day are depicted in Figure 6. The $OH + NO_2$ pathway shows no significe difference between clean and polluted day and dominate clean day nitrate formation potential. Since the level of OH and $NO_2$ less affected by the fine particle level. However, the rapid increase of $N_2O_5$ heterogeneous uptake pathway on polluted day is fatal and its peak formation rate at night over $OH + NO_2$ pathway which can be used to explain nighttime nitrate explosive growth.

As shown as Figure 6c, $OH + NO_2$ dominate nitrate production on clean day, while $N_2O_5$ uptake pathway only contributes 13.2 $\mu g\ m^{-3}$. On polluted days, the ability of $N_2O_5$ uptake grow fast which reached 25.3 $\mu g\ m^{-3}$, while OH pathway don't change too much. There is no distinct difference of daytime pathway ($OH + NO_2$) between clean day and polluted day, while nighttime pathway ratio rises from 38.4 % on clean day to 52.9 % on polluted day. The contribution of $N_2O_5$ hydrolysis on particular nitrate is vital at pollution condition.



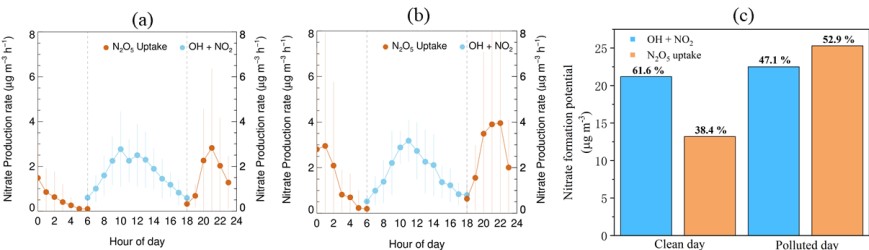

**Figure 6** The mean diurnal variations of nitrate production potential of clean day(a) and polluted day (b) and the P(NO$_3^-$) distribution of clean and polluted day (c).

### 3.5 Mitigation strategies of particulate nitrate and ozone productions

We selected two pollution episodes (Episode I (2019.05.30 00:00 - 2019.06.02 00:00) and IV (2019.06.14 17:30 - 2019.06.17 12:00)) to explore the mitigation way of ozone and nitrate pollution. Figure 7 shows the EKMA of P(O$_3$) and P(NO$_3^-$) of these two periods, O$_3$ located at VOCs controlling area in the two pollution episodes which consist with previous YRD urban ozone sensitivity study (Jiang et al., 2018; Zhang et al., 2020a; Xu et al., 2021). The best precursor reduction for O$_3$ is VOCs: NO$_x$ = 2:1 while nitrate located at transition area which means either of precursors reduction will mitigate nitrate pollution. For the regional and complex air pollution characteristics in this region, a fine particle-targeting reduction scheme will aggravate O$_3$ pollution. In contrast, the O$_3$-targeting scheme can mitigate O$_3$ and fine particle simultaneously.

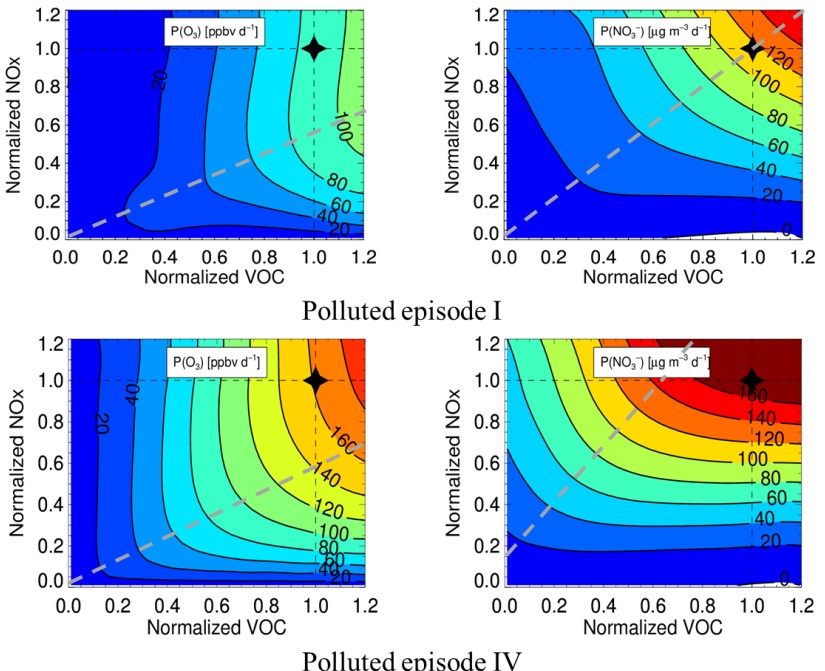

Polluted episode I

Polluted episode IV

**Figure 7** Isogram of P(O₃) and P(NO₃⁻) of polluted episode I (2019.05.30 00:00 - 2019.06.02 00:00) and IV (2019.06.14 17:30 - 2019.06.17 12:00) with different NOx and VOC reduction degree. Grey dash line represents the ridge line.

**4 Conclusion**

A comprehensive campaign was conducted to interpret the atmospheric oxidation
capacity and aerosol formation during May 30th to June 18th 2019 at Changzhou, China.
The high O₃ and PM₂.₅ concentration confirm complex air pollution characteristics in
Changzhou and nitrate accounts for 38.3 % mass concentration of water-soluble
components and 32.0 % total PM₂.₅. In addition, the average values of NOR are 0.32 in
PD, 0.25 in CD. The positive correlation between NOR and RH and inverse correlation
refer the contribution of N₂O₅ heterogenous uptake to nitrate formation.
Based on field observations of OH and related parameters, we show OH oxidation
of NO₂ pathway steadily contribute to nitrate formation no matter clean or polluted
period and domination clean day nitrate production (about 22 μg m⁻³). N₂O₅



heterogeneous uptake contribution grow rapidly on polluted day, from 13.2 μg m$^{-3}$
(38.4 %) in clean days to 25.3 μg m$^{-3}$ (52.9 %) in polluted days.

The precursor reduction simulation suggests the reduction ratio of VOCs:$NO_x$

equals 2:1 can simultaneously and effectively mitigate $O_3$ and fine particle pollution
during summertime complex pollution period in Changzhou. In order to more precisely
and delicately establish cooperative control scheme for regional $O_3$ and nitrate, the
regional and long-time filed campaigns are needed in the future, to analyze seasonal
and interannual variation of $O_3$ and nitrate and relevant parameters.

**Code/Data availability.** The datasets used in this study are available from the
corresponding author upon request (k.lu@pku.edu.cn).

**Author contributions.** K.D.L. and Y.H.Z. designed the study. T.Y.Z analyzed the data
and wrote the paper with input from all authors.

**Competing interests**. The authors declare that they have no conflicts of interest.

**Acknowledgments**. This project is supported by the National Natural Science
Foundation of China (21976006); the Beijing Municipal Natural Science Foundation
for Distinguished Young Scholars (JQ19031); the National Research Program for Key
Issue in Air Pollution Control (DQGG0103-01, 2019YFC0214800). Thanks for the data
contributed by field campaign team.

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
