# Peer review of "Elucidate the Formation Mechanism of Particulate"

_Atmospheric Chemistry and Physics, 2022_

## Referee Comment (RC2)

"Elucidate the Formation Mechanism of Particulate Nitrate Based on Direct Radical Observations in Yangtze River Delta summer 2019" by Zhai et al. conducted field measurements from May to June, 2019 at regional site in Changzhou, Jiangsu province in China. Authors continuously measured the nitrate, OH radical, N2O5, NO2, O3, and relative meteorological parameters to shed light on the particulate nitrate formation mechanisms. In the model simulation, the authors focused on the OH+NO2 and heterogeneous uptake of N2O5 process, it is not clear if the author investigated the contribution of heterogeneous NO2 uptake and its hydrolysis in forming nitrate. I recommend publication of this paper in Atmospheric Chemistry and Physics after addressing the following questions.

Line 75-77:

The author stated that NO2 heterogeneous uptake is a minor pathway based on previous studies. Have the authors incorporated heterogeneous NO2 uptake in the model? The previous model works reported that the mean contribution of heterogeneous NO2 to nitrate formation is 6.3% ~19.0%, which increases to ~35.9% during extreme haze events (Chan et al., 2021; Qiu et al., 2019). From my perspective, heterogeneous NO2 uptake can be an important process in a certain condition.

Line 255:

What does the "water-soluble ion" refer to?

Line 275-276:

Overall, NOR increases as the RH increases, which may indicate the liquid water content play a crucial role in nitrate formation. However, as I mentioned earlier, heterogeneous NO2 uptake and subsequent NO2 hydrolysis also rely on the high liquid water content. It is possible that heterogeneous NO2 is also essential in forming nitrate. It would be better to incorporate this process in the model to evaluate its importance in the atmosphere further.

Line 282:

Can the author elaborate more on the NOR difference between PD and CD? From my perspective, particular factors should lead to such a discrepancy. Any suggestions?

Line 343:

Any suggestions on the increased ability of N2O5 uptake on a polluted day? Is it due to the different composition of particles on PD and CD? Need more elaboration.

Reference:

(1) Chan, Y.C.; Evans, M.J.; He, P.; Holmes, C.D.; Jaegl´e, L.; Kasibhatla, P.; Liu, X.Y.; Sherwen, T.; Thornton, J.A.; Wang, X.; Xie, Z.; Zhai, S.; Alexander, B. Heterogeneous Nitrate Production Mechanisms in Intense Haze Events in the North China Plain. J Geophys Res Atmos 2021;126:e2021JD034688.

(2) Qiu, X., Ying, Q., Wang, S., Duan, L., Zhao, J., Xing, J., Ding, D., Sun, Y., Liu, B., Shi, A., Yan, X., Xu, Q., Hao, J., 2019. Modeling the impact of heterogeneous reactions of chlorine on summertime nitrate formation in Beijing. China. Atmos Chem Phys 19, 6737–6747.

---

## Author Comment (AC1)

Dear Editor and Referee

Thank you for reviewing and commenting upon our manuscript, "Elucidate the Formation Mechanism of Particulate Nitrate Based on Direct Radical Observations in Yangtze River Delta summer 2019" by Tianyu Zhai et al., Atmos. Chem. Phys. Discuss., https://doi.org/10.5194/acp-2022-548-RC1, 2022. As detailed below, the reviewer's comments are in italicized font, and our responses to the comments are in regular font. New or modified text is in blue.

We've responded to each comment individually below and would like to draw your attention to two major concerns:

Comment 1:
*Using a constant coefficient for the whole campaign seems to be less convincing, although 0.035 was a reasonable value in this area. I suggest to perform some uncertainty tests or at least choose different coefficient for clean days and polluted days, respectively, as the aerosol composition and water content were not supposed to be the same.*

Reply: We thank for these constructive comments and suggestions to improve the quality of our manuscript. In the article, we add the timeseries and box-whisker plots of $\gamma\_p$ for clean days and polluted days, respectively (Figure 6). The average $\gamma\_P$ is 0.069 ± 0.0050 in polluted condition and 0.0036 ± 0.0026 in clean condition. Moreover, the different values $\gamma\_p$ will be applied to subsequent nitrate formation contribution calculation. More clarifications have been added in section 3.3 as follows:

The other approach is the parameterization by (Yu et al., 2020) which depicted as follows:

$$\gamma\_P = \frac{4}{c}\frac{V_a}{S_a} \ K_H \times 3.0 \times 10^4 \times [H_2O] \left( 1 - \frac{1}{\left(0.033 \times \frac{[H_2O]}{[NO_3^-]}\right) + 1 + \left(3.4 \times \frac{[Cl^-]}{[NO_3^-]}\right)} \right) \quad \text{Eq11}$$

where $V_a/S_a$ is the measured aerosol volume to surface area ratio by SMPS; $K_H$ is Henry's law coefficient which is set as 51 as recommended; $[NO_3^-]$ and $[Cl^-]$ are aerosol inorganic concentration measured by Marga; $[H_2O]$ is aerosol water content calculated through ISORROPIA II. The valid parameterization calculated $N_2O_5$ uptake

coefficient (note as $\gamma\_P$) from May 30th to June 08th, 2019 shows in Figure 6, there is a good consistency between the trends of $\gamma\_P$ and aerosol water content. Nighttime $\gamma\_P$ varys from 0.001 to 0.024 with average 0.069 ± 0.0050 on polluted condition and 0.0036 ± 0.0026 on clean condition. the $N_2O_5$ uptake coefficient shows good correlation to RH and aerosol water content. For $N_2O_5$ uptake coefficient, although particulate nitrate mass concentration increased during pollution event, antagonistic effect on $N_2O_5$ uptake coefficient was not obvious for the nitrate molarity decreasing.

Furthermore, we compare the difference between $\gamma\_S$ and $\gamma\_P$ [h]. Taking the night of May 30th as example, the $\gamma\_S$ is 0.089 while $\gamma\_P$ ranges from 0.024 to 0.057 with average value as 0.013 ± 0.0051. The difference between steady-state and parameterization is significant, one possible explain is uncertainty for stationary-state approximation caused by local NO or VOCs emission (Brown et al., 2009; Chen et al., 2022). Another reason is parameterization by Yu et al. ignore the impact from organic matter on fine particle. The difference aerosol composition between this work and Yu et al may also bring uncertainty. Overall consideration, $\gamma\_P$ will be chosen for $N_2O_5$ heterogeneous uptake coefficient in later analysis and discussion.

[Figure]

**Figure 6** Results of $N_2O_5$ uptake coefficients through parameterization ($\gamma\_P$). (a) shows timeseries of $\gamma\_P$ and ISORROPIA II results of aerosol water content (AWC). (b) is the box-plot of $\gamma\_P$ on polluted day and clean day, hollow square represents the mean value and the solid line across the box shows the median score for the data set, while the top and bottom whiskers represent 90 % and 10 % value of $\gamma\_P$, respectively.

Comment 2: *Please explain the reason for the significant difference between gamma_s and gamma_p.*

Reply: First, local NO or VOCs emission during nighttime will break steady chemical condition for stationary-state approximation as the temperature and $kNO_3$ meet requirements reported in the literature (Brown et al., 2009; Chen et al., 2022). What's more, the ignorance of organic matter influence on $N_2O_5$ heterogeneous uptake coefficient in parameterization will also bring uncertainty.

Word and format problems have been corrected as suggested for other specific comments.

Line 91 change "impact factor" to "controlling factors"

Line 241 change 'appeal" to "appear"

The conclusion at lines 276~279 has been deleted for no more evidence.

Thank you again for your thoughtful comments.

Reference

Brown, S. S., Dube, W. P., Fuchs, H., Ryerson, T. B., Wollny, A. G., Brock, C. A., Bahreini, R., Middlebrook, A. M., Neuman, J. A., Atlas, E., Roberts, J. M., Osthoff, H. D., Trainer, M., Fehsenfeld, F. C., and Ravishankara, A. R.: Reactive uptake coefficients for N2O5 determined from aircraft measurements during the Second Texas Air Quality Study: Comparison to current model parameterizations, Journal of Geophysical Research-Atmospheres, 114, 10.1029/2008jd011679, 2009.

Chen, X., Wang, H., and Lu, K.: Interpretation of NO3-N2O5 observation via steady state in high-aerosol air mass: the impact of equilibrium coefficient in ambient conditions, Atmospheric Chemistry and Physics, 22, 3525-3533, 10.5194/acp-22-3525-2022, 2022.

Yu, C., Wang, Z., Xia, M., Fu, X., Wang, W. H., Tham, Y. J., Chen, T. S., Zheng, P. G., Li, H. Y., Shan, Y., Wang, X. F., Xue, L. K., Zhou, Y., Yue, D. L., Ou, Y. B., Gao, J., Lu, K. D., Brown, S. S., Zhang, Y. H., and Wang, T.: Heterogeneous N2O5 reactions on atmospheric aerosols at four Chinese sites: improving model representation of uptake parameters, Atmospheric Chemistry and Physics, 20, 4367-4378, 10.5194/acp-20-4367-2020, 2020.

---

## Author Comment (AC2)

Dear Editor and Referee

Thank you for reviewing and commenting upon our manuscript, "Elucidate the Formation Mechanism of Particulate Nitrate Based on Direct Radical Observations in Yangtze River Delta summer 2019" by Tianyu Zhai et al., Atmos. Chem. Phys. Discuss., https://doi.org/10.5194/acp-2022-548-RC1, 2022. As detailed below, the reviewer's comments are in italicized font, and our responses to the comments are in regular font. New or modified text is in blue.

We've responded to each comment individually below and would like to draw your attention to two major concerns:

*Comment 1: About Line 75-77 and Line 275-276, the concern about the contribution of $NO_2$ heterogeneous uptake.*

Reply: Thanks for this precious advice. First, we change the description of $NO_2$ heterogeneous uptake contribution on nitrate formation and add relative literature reports around line 75. What's more, the $NO_2$ heterogeneous uptake pathway analysis has been added to chapter 3.4. The $NO_2$ uptake coefficient is set as $5.8 \times 10^{-6}$ depending on the report by Yu et al. (2021), which is the result of 70% RH on urban grime. The $NO_2$ heterogeneous uptake pathway contribution on nitrate formation mass increased twice on polluted day than on clean day. However, there remains a distance among the value of $NO_2$ heterogeneous uptake, OH+$NO_2$, and $N_2O_5$ heterogeneous uptake. More clarifications have been added in section 3.3 as follows:

As shown in Figure 7c, OH + $NO_2$ dominates nitrate production on clean day, while the $N_2O_5$ uptake pathway only contributes 13.6 μg m$^{-3}$. On polluted days, the ability of $N_2O_5$ uptake grows fast which reached 50.1 μg m$^{-3}$, while OH pathway doesn't change too much. There is no distinct difference of daytime pathway (OH + $NO_2$) between clean day and polluted day, while the nighttime pathway ratio rises from 38.1 % on clean day to 67.2 % on polluted day.$NO_2$ heterogeneous uptake increases from 0.93 μg m$^{-3}$ on clean day to 2.0 μg m$^{-3}$ on polluted day, but the contribution proportion do not change obviously. Both the higher $N_2O_5$ uptake coefficient and higher $S_a$ on polluted day increase the contribution of $N_2O_5$ hydrolysis on particular nitrate at pollution

condition.

[Figure]

**Figure 7** The mean diurnal variations of nitrate production potential of clean day(a) and polluted day (b) and the $P(NO_3^-)$ distribution of clean and polluted day (c).

*Comment 2: What does the "water-soluble ion" refer to?*

Reply: "water-soluble ion" refer to $Na^+$, $K^+$, $Ca^{2+}$, $Mg^{2+}$, $NH_4^+$, $NO_3^-$, $Cl^-$ and $SO_4^{2-}$ components in particle. In order to avoid confusion, "water-soluble particulate components" is used to replace all the "water-soluble ion".

*Comment 3: Can the author elaborate more on the NOR difference between PD and CD? From my perspective, particular factors should lead to such a discrepancy. Any suggestions?*

Reply: The NOR increased during PD reveals the fast transformation of $NO_2$ to $NO_3^-$, which is in accordance with nitrate ratio explosive growth during PD. As discussed in chapter 3.4, both the $N_2O_5$ heterogeneous uptake pathway and $NO_2$ heterogeneous uptake pathway increased more than twice during PD. In our opinion, there is no significant difference in the RH, $NO_2$ concentration, and $N_2O_5$ concentration between PD and CD. Even the $N_2O_5$ concentration decreased during PD night. The nitrate formation contribution increase is controlled by the aerosol surface growth and aerosol water content increase which is due to the change of particulate composition.

*Comment 4: Any suggestions on the increased ability of $N_2O_5$ uptake on a polluted day? Is it*

*due to the different composition of particles on PD and CD? Need more elaboration.*

Reply: Both the higher $N_2O_5$ uptake coefficient and higher $S_a$ on polluted day increase the contribution of $N_2O_5$ hydrolysis on particular nitrate at pollution condition. Both the $N_2O_5$ uptake coefficient and $S_a$ shows a good correlation to RH and aerosol water content. For the $N_2O_5$ uptake coefficient, although particulate nitrate mass concentration increased during pollution event, antagonistic effect on $N_2O_5$ uptake coefficient was not obvious for the nitrate molarity decreasing.

Thank you again for your thoughtful comments.

Reference

Yu, C. A., Wang, Z., Ma, Q. X., Xue, L. K., George, C., and Wang, T.: Measurement of heterogeneous uptake of NO2 on inorganic particles, sea water and urban grime, J. Environ. Sci., 106, 124-135, 10.1016/j.jes.2021.01.018, 2021.